

# Reference maps of soil phosphorus for the pan-Amazon region

João P. Darela-Filho[1,2,3], Anja Rammig[3], Katrin Fleischer[3,5], Tatiana Reichert[3], Laynara F. Lugli[3], Carlos Alberto Quesada[6], Luis Carlos Colocho Hurtarte[4,8], Mateus Dantas de Paula[7], and David M. Lapola[2]

[1]São Paulo State University (Unesp), Institute of Biosciences, Rio Claro, 13506-900, Brazil
[2]University of Campinas (Unicamp) Center for Meteorological and Climatic Research Applied to Agriculture (CEPAGRI), Earth System Science Laboratory (LabTerra), Campinas – SP,13083-886, Brazil
[3]Technical University of Munich (TUM), School of Life Sciences, Freising, 85354, Germany
[4]European Synchrotron Radiation Facility, Beamline ID21, Grenoble 38100, France
[5]Max-Planck-Institute for Biogeochemistry, Department of Biogeochemical Signals, Jena, 07745, Germany
[6]National Institute for Amazonian Research – INPA. Avenida André Araújo, 2236, Manaus, Amazonas, 69060-001, Brazil
[7]Senckenberg Biodiversity and Climate Research Centre (SBiK-F), Frankfurt am Main 60325, Germany
[8]Diamond Light Source Ltd., Didcot, Oxfordshire OX11 0DE, UK

*Correspondence to*: João Paulo Darela-Filho (darelafilho@gmail.com)

**Abstract.** Phosphorus (P) is recognized as an important driver of terrestrial primary productivity across biomes. Several recent
developments in process-based vegetation models aim at the concomitant representation of the carbon (C), nitrogen (N) and P cycles in terrestrial ecosystems, building upon the ecological stoichiometry and the processes that govern nutrient availability in soils. Thus, understanding the spatial distribution of P forms in soil is fundamental to initialize and/or evaluate process-based models that include the biogeochemical cycle of P. One of the major constraints for the large-scale application of these models is the lack of data related to the spatial patterns of the various forms of P present in soils, given the sparse nature of in
situ observations. We applied a model selection approach based on random forest regressions models trained and tested for the prediction of different P forms (total, available, organic, and inorganic P) – obtained by the Hedley sequential extraction method. As input for the models, reference soil group and textural properties, geolocation, N and C contents, terrain elevation and slope, soil pH and mean annual precipitation and temperature from 108 sites of the RAINFOR network were used. The selected models were then applied to predict the target P forms using several spatially explicit datasets containing contiguous
estimated values  across the area of interest. Here, we present a set of maps depicting the distribution of total, available, organic, and inorganic P forms in the topsoil profile (0 - 30 cm) of the pan-Amazon region in the spatial resolution of 0.5 x 0.5 degrees. The random forest regression models presented a good level of mean accuracy for the total, available, organic, and inorganic P forms (77.37 %, 76,86 %, 75.14 %, and 68.23 % respectively). Our results confirm that the mapped area have generally very low total P concentration status with a clear gradient of soil development and nutrient content. Total N was the most important
variable for the prediction of all target P forms and the analysis of partial dependence indicates several features that are also related with soil concentration of all target P forms. We observed that gaps in the data used to train and test the random forest models, especially in the most elevated areas, constitute a problem to the methods applied here. However, most of the area could be mapped with a good level of accuracy. Also, the biases of gridded data used for model prediction are introduced in the P maps. Nonetheless, the final map of total P resembles the expected geographical patterns. Our maps may be useful for





the parametrization and evaluation of process-based terrestrial ecosystem models as well as other types of models. Also, they can promote the testing of new hypothesis about the gradient and status of P availability and soil-vegetation feedbacks in the pan-Amazon region.

**1 Introduction**

Phosphorus (P) is one of the main plant macronutrients and it is known to pose major limitations on the terrestrial primary

productivity in the tropics (Wollast et al., 1993, Du et al., 2020; Cunha et al., 2022). In ecosystems with highly weathered soils, as in the pan-Amazon region (RAISG, 2022), the processes that govern the local readily available P (orthophosphates, i.e., salts and esters of the orthophosphoric acid) in soils are largely related to the mineralization of organic matter mediated by decomposers (da Silva et al., 2022). In such weathered soils, the readily available P for living organisms, including plants, is only a small fraction of the total P (Quesada et al., 2010; Vitousek et al., 2010; Walker and Syers, 1976), whereas most of

the P is chemically bound or adsorbed to other molecules, forming stable organic and inorganic compounds (Buendia et al., 2014; Smeck, 1985). Plants and microorganisms developed several strategies to overcome the lack of available P in these soils, explore less available P pools (Lugli et al., 2020; Reichert et al., 2022; Lambers, 2022 and citations therein) and preserve the acquired P, resulting in a tight cycling and very small leaching (Wilcke et al., 2019).

The different P pools vary in time and respond differentially to environmental conditions; thus, they can act as sinks or sources

of available P (Schubert et al. 2020; Helfenstein et al., 2018; Gama-Rodrigues et al., 2014), and the understanding of the processes that govern the cycling of P in these pools is important to the models that aim at simulating the productivity of terrestrial ecosystems. Dynamic Global Vegetation Models (DGVM) that include the P cycle often rely on maps of soil P for model benchmarking or model initialization. These maps are built upon models of varying complexity that link soil P pools to variables like soil type, soil age, lithology, soil C and N content and soil texture (*e.g.*, Castanho et al., 2013; Goll et al., 2012;

Wang et al., 2010). The global maps of soil P forms created and described by Yang et al. (2013) (available in Yang et al., 2014) were derived from several global soil datasets combined with the current scientific understanding of P transformations during pedogenesis. In this approach, total P content was obtained from estimates of initial rock P content and model-based P loss rates from soil chronosequences. The attribution of each P fraction, from the estimated total P, is based on averaged values of each fraction extracted sequentially in different soil reference groups (Yang et al., 2013). The low number of soil P

measurements and the lack of knowledge about the processes controlling P during pedogenesis are the sources of a high level of uncertainty in these global maps (Yang and Post, 2011). Yet, some studies provided evidence that edaphic and climatic factors (He et al., 2021, Gama-Rodrigues et al., 2014) other than reference soil groups can be used to predict the size of the P pools in soils.

Thus, we here aim to overcome the low number of soil P measurements for certain reference soil groups developing a set of

maps of the different P forms for the pan-Amazon region. Firstly, we used a model selection approach to fit, test and evaluate a set of machine learning regression models using 108 sites with measurements of sequential extraction of soil P and a set of





environmental variables that are complementary to the reference soil groups. Finally, we used the selected models to predict the spatial distributions of soil P forms using gridded datasets of the same environmental variables used to fit the random forest regression models.

## 2 Material and methods

### 2.1 Phosphorus data: the fitting dataset

The soil and other environmental measurements at 92 sites selected (Hou et al., 2018) for this study originally presented in the studies of Quesada et al. (2010) and Lloyd et al. (2015). The soil P measurements and analysis are described and standardized by Quesada et al. (2010) and refer to the 0-30 cm depth of the soil profile. Additionally, we included here sixteen unpublished site data from the study of Quesada et al. (2020), resulting in a total of 108 observations (Fig. 1 and S6). All soil samples were collected under the RAINFOR protocol for soil sampling (RAINFOR, 2022). The soil samples were submitted to a sequential P extraction by Hedley fractionation procedure (Carter and Gregorich, 2008; Hedley and Stewart, 1982; Tiessen and Moir, 1993). In this dataset, the P fractionation data is complemented by several variables. For this study, soil texture (sand, silt, and clay fraction), C and N content, pH, elevation, soil reference group, mean annual precipitation and temperature, latitude, and longitude were used (Table 1). The calculations of soil attributes, like pH, texture, and C and N contents are described in Quesada et al. (2010). Elevation and climatic variables were extracted by Quesada et al. (2010, 2020) and Lloyd et al. (2015) from a digital elevation model (Shuttle Radar Topography Mission database - Digital Elevation Model – SRTM-DEM) and the WordClim dataset (Fick and Hijmans, 2017), respectively. The slope for all sites was calculated using the same digital elevation model (Saatchi, 2013) with 30 arc-seconds of resolution. The categorical variable reference soil group was transformed into a set of binary variables via one-hot encoding to train the random forest regression models. The process of one-hot encoding originated 16 binary variables, being one for each reference soil group.

Due to the complex forms in which soil P occurs in soils, sequential chemical extraction methods are commonly used (Hedley and Stewart, 1982; Tiessen and Moir, 1993) resulting in groupings of "ecosystem-relevant" pools (Gama-Rodrigues et al., 2014; Hou et al., 2018), which we here refer to as P forms. There is ongoing discussion about the interpretation of the P fractions obtained via the sequential extraction methods and no consensus on how to organize this into reservoirs in the ecosystem (Gu and Margenot, 2020; Barrow et al., 2020). Nonetheless, up to now this is the most commonly used method to determine soil P fractions, and in this study, we considered five P pools or forms based on previous works (Hedley et al., 1982; Yang et al., 2013; Hou et al., 2018).

(1) The orthophosphates, and other inorganic forms which can be easily converted into orthophosphates with residence times that vary from minutes to hours (Helfenstein et al. 2020) are classified as available P. This P pool is composed by the Resin and $NaHCO_3$ (sodium bicarbonate) inorganic fractions derived from the sequential extraction process. (2) The forms of P resulting from chemical bonds with inorganic compounds that are less, but still accessible by plants – i.e., have mean residence times that vary from days to months (Helfenstein et al. 2020) are classified here as inorganic P; this P pool is composed by the



NaOH (Sodium hydroxide) inorganic P fraction derived from the sequential extraction method. (3) All the fractions of P related
to the organic matter are classified as organic P; this pool is composed of the organic fractions extracted with NaHCO$_3$ and
NaOH in the sequential extraction process.The inorganic and organic P pools are accessible by plants via alternative acquisition
and mobilization strategies (Lambers, 2022). (4) P linked to primary minerals is the form that is present in the parent lithology.
This form of soil P is commonly named as primary mineral P and is transformed by weathering, which liberates inorganic P
in the soil. Here we assume that the primary mineral P is composed of the fraction extracted with HCl (Hydrochloric acid -
Calcium bound P) in the sequential fractionation process. The mean residence time of primary mineral P fall in timescales that
vary from years to millennia (Helfenstein et al. 2020). (5) There are less accessible forms of P that are tightly bonded or
adsorbed to other molecules in soil, organic or inorganic, and due to the strength of these chemical linkages, only become
available to plants at longer time scales and/or with higher costs for mobilization and acquisition (Schubert et al. 2020). These
forms are grouped in the pool of occluded P and are represented here by the residual P obtained by Quesada et al. (2010). This
residual pool is estimated from the subtraction between total P (see next paragraph) and the sum of all preceding forms (1 -
4). Both occluded and primary mineral P are formed by stable compounds that have residence times ranging from years to
millennia (Lambers, 2022; Helfenstein et al., 2018).

The total P pool comprises all the forms of P described above and is the total P extracted with acid digestion using a
concentrated solution of H$_2$SO$_4$ followed by H$_2$O$_2$ in replicate samples to avoid errors caused by the laboratory procedures
(Quesada et al., 2010). The primary mineral P and the occluded P forms were excluded from this analysis after a preliminary
fitting of random forest regressions. The models fitted for these two pools received low values of accuracy and predicted highly
overestimated values in some specific regions. We interpret that the information from the set of variables in Table 1 is therefore
insufficient to generate predictions of the primary and occluded P forms. We speculate that because these fractions are more
linked to geochemical drivers such as origin of parent rock material, weathering rates, soil age and, Al and Fe contents these
could be good predictors to test for in future studies. To address this issue, we estimated the aggregated size of the occluded P
and primary P pools, by subtracting the available, organic and inorganic P forms from the estimated total P. The aggregation
of P fractions extracted sequentially into P pools is summarized in Table 2. It is important to note that the total, available,
organic, and inorganic P forms described in this section were used only as targets – i.e., response variables, for model fitting
purposes. The complete description of the use of the phosphorus dataset for model fitting is described in Sect. 2.3.

**2.2 Predictive variables for constructing the P maps: the predictive dataset**

The predictor data is composed of a set of five distinct spatially explicit datasets in raster and vector formats. Spatial and
numerical transformations were applied to the original datasets to fit the purpose of this study. The SOTERLAC (Soil and
Terrain database for the Latin America and Caribe) vector data of soil types (Dijkshoorn et al., 2005) for the study area were
rasterized, respecting the original scale of 1: 5.000.000, to 30 arc-seconds of lat-lon resolution. The major WRB (World
Reference Base for soil resources) reference soil groups were attributed following Quesada et al. (2010). Mean annual
precipitation and temperature (MAP, °C and MAT, mm/year), were extracted from the WorldClim database (Fick and Hijmans,



2017) and refer to the climatological means for the 2001-2010 decade. Soil pH in water, soil texture (sand, silt, and clay, %), and total organic C (TOC, %) were extracted from the Harmonized World Soil Database – HWSD version 1.2 (Wieder, 2014) for the topsoil (0-30 cm). Soil total N (TN, %) for the topsoil, was extracted from the - IGBP-DIS dataset (Global Soil Data

Task, 2014). The elevation data was extracted from the SRTM-DEM of the Latin America (Saatchi, 2013, Farr et al. 2007 ). Terrain slope (%) was estimated from the elevation data. All spatial raster data sets were aggregated to the 0.5° x 0.5° resolution. The reference soil group map was aggregated to the final resolution using the modal value, and the remaining datasets were aggregated using the mean values. The categorical variable reference soil group was transformed into a set of binary variables using one-hot encoding before the application of the trained random forest models to the prediction of the

phosphorus maps. Less common reference soil groups found in the region were excluded during the aggregation of the raster datasets (Table S4). Leptosols, a common soil reference group in the elevated areas in the Andes and the northern and southern crystalline shields, were not sampled, i.e., were not in the fitting dataset. These were considered undefined and received the value zero or false to all sixteen binary variables representing the reference soil group in the predictive dataset (Table S4).

To identify the areas – i.e., grid cells, where the predictor data presents high multivariate dissimilarity with the observed data,

such that the predictive power of the random forest models can be compromised, we calculated the Dissimilarity Index (DI, Meyer and Pebesma, 2021). The DI denotes the multidimensional dissimilarity of a given predictor grid cell (a spatial grain composed of the data described in this section) in relation to the fitting dataset (the phosphorus dataset, described in the previous section). The DI can assume values equal to or greater than zero. Low  metric values indicate that the given predictor grid cell presents low dissimilarity in relation to the observations in the fitting dataset. We excluded from the figures all grid

cells with DI values above the sum of the third quartile with the inter-quartile range (Fig. S2).

## 2.3 Random Forest models

We employed a model selection approach based on model regressions using the random forest algorithm (Breiman, 2001; Cutler et al., 2012). For each target P form (available, organic, inorganic, and total P), $10^5$ random forest regression models were trained and tested using the fitting dataset (Sect. 2.1). We used the Scikit-Learn *RandomForestRegressor* (Pedregosa et

al., 2011). For each random forest model being fitted, 100 decision tree estimators were used. The phosphorus dataset was randomly split into training and test data (75% and 25% of the samples, respectively). During the training and testing, every fitted model and its relative dataset split were assigned to a random state, represented by an integer between 0 and 99999. Thus, we were able to identify each model and associate it with the division of the phosphorus dataset for training and testing. As criterions for the selection of random forest models, we adopted an accuracy measure (Eq. 1) based on the Mean Absolute

Percentage Error (MAPE, De Myttenaere et al., 2016) and a Monte Carlo cross-validation procedure (Stone, 1974; Kuhn and Johnson, 2013) in which the models were cross-validated on fifteen random splits of the phosphorus dataset, using the same ratio of the fitting dataset splitting as in the training phase. The metric used to evaluate the model's performance in the cross-validation phase was the coefficient of determination ($R^2$). For each target variable – i.e., P form, we selected the models with accuracy and cross-validation $R^2$ scores above arbitrarily chosen threshold values based on preliminary evaluations of a



thousand random forest regression models for each target variable. The chosen threshold values (Table 3) were defined to have a minimal number of models after the training of the $10^5$ models for each target variable. The exclusion of the primary mineral and occluded P forms were also based on these preliminary tests. The selection criteria, the number of selected models for each target variable, the mean accuracy (Aµ), and cross-validation $R^2$ of the selected models are presented in Table 3. The model accuracy is defined as:

$$Accuracy = 100 - MAPE \text{ (\%)} \tag{1}$$

We calculated two additional model evaluation metrics. The mean absolute error and the coefficient of determination (not to be confused with the cross-validation $R^2$) for each selected model for all P forms. To estimate the importance of the features, we calculated the Mean Decrease in Accuracy (MDA) for each selected model. The MDA is calculated via permutation where a given selected model is tested several times (120 in our case) with rearrangements (shuffling of a single feature) in the model's respective testing split of the fitting dataset. These rearrangements aim to eliminate the potential relationships between the permutated features and the target variable and estimate how much accuracy is lost in the process. Thus, for a given model, features that show higher values of MDA provide higher predictive power.

To identify the effects of the most important input features on the predictions of the random forest models, we calculated the partial dependence (Hastie et al., 2001) and plotted the individual conditional expectation (Goldstein et al., 2014) for the most accurate random forest model selected for each P form. To generate the final maps of the P forms we grouped the maps predicted by the selected models for each P form via mean and calculate the standard error (both in mg kg$^{-1}$). We used the topsoil bulk density aggregated from the HWSD-v1.2 dataset to calculate the stocks of each P form in the topsoil (in Pg – petagrams). The steps involved in model training, testing, and selection, and the selected models used for the prediction of P forms are summarized in Fig. S1. We compared our final map of total P concentration with the map of He at al. (2021). The observed values of total P in the fitting dataset were compared with the respective predicted values in the corresponding gridcells of the estimated maps via the pearson correlation coefficient. The maps presented here were coloured using the scientific colour maps version 8.0 (Crameri, 2021; Crameri et al., 2020).

## 3 Results

### 3.1 Descriptive statistics of the datasets

The mean concentration of total P in the fitting dataset across the 108 sites was 284.13 mg kg$^{-1}$ (Table S1). The primary mineral and occluded forms of P corresponded to 53.61 % of the mean total P concentration, followed by the organic P (27.78 %), inorganic P (11.9 %) and lastly by the available P (6.71 %). The descriptive statistics of the features in the fitting dataset are presented in Table S2. The predictive dataset was obtained after the adequation of several well-known spatial datasets covering the area of interest. The descriptive statistics of the features in the predictive dataset are presented in Table S3. The use of the dissimilarity index (DI; Fig. S2) resulted in the exclusion of several grid cells from the predictive dataset (hatched





areas in Fig 1-4). The area covered by the predictive dataset is approximately 8.4 x $10^6$ km$^2$; of which 8.2 % was excluded for the prediction of the total P map. The excluded areas for the final maps of the available, organic, and inorganic P forms were 11.6%, 11.4 %, and 9.3 %, respectively. The excluded areas for each P form overlap in several locations. The intersection of all excluded areas represented 16.3 % of the covered area in the predictive dataset (hatched areas in Fig. 5). The excluded grid

cells in the DI analysis have higher values of elevation, slope, TOC and TN and lower values of MAT when compared with the non-excluded grid cells (Fig. S3). The predicted values of P forms were consistently higher in the excluded areas (Fig. S4). After the exclusion of the grid cells with high DI values, the distribution of the features in the predictive dataset fell approximately within the distributions of the features in the P dataset (Fig. S5). The descriptive statistics of the predictive dataset without the exclusion of the grid cells with high DI values can be found in table S3.

**3.2 Estimated P stocks in the topsoil and the geographical patterns of P pools in the study area**

All estimates presented in this section do not consider the excluded areas in the DI analysis. The estimated P form concentrations (Fig.1 - 4) are the mean values for each grid cell as predicted by the selected models. The mean values are presented with the standard error. The estimated stock of total P (Fig. 1) is 0.61 Pg P with mean concentration of 197.29 ranging from 58.37 to 486 mg kg$^{-1}$ in the topsoil profile. We found total P concentration above the mean values in the

Amazonian foreland basins; and in the central area of the western portion of the Amazon rift (the region that divides the Brazilian and Guiana shields), which corresponds approximately to the catchments of the Solimões, Juruá, and Purús Rivers at the center of the study area (Fig. S6). In comparison, the mean total P concentration found in the He et al. (2021) map, over the same area, is 336.6 mg kg$^{-1}$, ranging from 191.2 to 961.1 mg kg$^{-1}$. The estimated stock of available P (Fig. 2) is 0.04 Pg P – or 5.7 % of the total P stock, and the estimated mean concentration is 11.74 mg kg$^{-1}$, ranging from 6.22 to 62.62 mg kg$^{-1}$.

The concentration of available P is lower than the mean in most of the central and north portions of the study area, but higher than the mean in the regions under the influence of the orogenic systems in the western, southern, and northern regions,characterized by elevations higher than 600 m (Fig. SM6). The stock of organic P (Fig. 3) is 0.13 Pg P (21 % of the total P stock). The estimated mean concentration of organic P ranges from 15.98 to 127.35 mg kg$^{-1}$ with mean values of 44.25 mg kg$^{-1}$. The estimated stock of inorganic P (Fig. 4) is 0.06 Pg P (9.83 % of the estimated total P stock) with a mean

concentration of 21.68 mg kg$^{-1}$, ranging from 5.13 to 63.16 mg kg$^{-1}$. The spatial patterns of organic and inorganic P concentrations follow the spatial pattern of the total P concentrations. The estimated stocks of primary mineral and occluded P correspond to 66 % (0.4 Pg P) of the predicted stock of total P, with a mean concentration value of 114.4 mg kg$^{-1}$ ranging from 15.32 to 263.33 mg kg$^{-1}$ (Fig. 5).

**3.3 Model performance and relations between predictive features and target P forms**

For each modelled P form (total, available, organic, and inorganic P) several models were selected based on minimum thresholds of accuracy (Eq. 1) and performance in a Monte-Carlo cross-validation procedure. Both evaluation metrics were

Earth System
Science
Data

Open Access | Discussions

calculated during the training and testing of the random forest regression models with the P (fitting) dataset. We selected 300 models for the prediction of the total P concentration maps (Table 3). The models presented a mean accuracy of 77.37 % (Fig. 1, left panel) with a mean score ($R^2$) of 0.7 in the cross-validation (Fig. S7). The importance score (Fig. 6) shows the features

that confer more valuable information and that increase the accuracy of the random forest models in the training/testing phase. Total N has high values of MDA for all target P forms and highest values for total P. Total N, pH, sand, mean annual temperature, silt, and total organic C were the most important predictive features for the models fitted with total P as target (Fig. 6). Total P presented positive non-linear relations with total N, silt, and total organic C; a non-monotonic relation with pH, and negative non-linear relations with MAT and sand fraction as shown by the partial dependency plots (Fig. S8).

The 419 models selected for prediction of the available P (Table 3) presented a mean accuracy of 76.85 % with a score of 0.6 in the cross-validation (Figs. 2 and S9). The variables with higher values of MDA were total N, MAP, total organic C, pH, elevation, and slope (Fig. 6). All the listed variables presented positive non-linear relations with available P apart from MAP, that showed a non-linear relation (Fig. S9). For the prediction of organic P, 247 random forest regression models were selected (Table 3) with a mean accuracy of 75.14 % and cross-validation $R^2$ of 0.67 (Figs. 3 and S10). The variables with higher values

of MDA for organic P (Fig. 6), on average, were total N, MAT, silt, elevation, total organic C, and pH. All variables presented positive relations with organic P concentration except MAT (Fig. S12). For the inorganic P form, 102 models with a mean accuracy of 68.23 % and a cross-validation score of 0.58 were selected (Figs. 4 and S13). The variables with higher values of MDA for the inorganic P form were total N, sand, total organic C, MAT, latitude, and clay (Fig. 6). Total N, total organic C, and clay presented positive relations with inorganic P, in contrast, MAT and sand showed negative relations (Fig. S14). In our

analysis, the reference soil group has very low values of MDA, indicating that they are not powering the accuracy of the random forest models and, thus, are not good predictors for P forms in the soil in our statistical approach. The predicted values of total P in our map presented a correlation coefficient of 0.64 (p < 0.01) when compared with the observed values in the fitting dataset. For comparison, the map of total P of He et al. (2021) presented a correlation coefficient of 0.35 (p < 0.01).

## 4 Discussion

We used soil data sampled in 108 plots of the RAINFOR network in the pan-Amazon region (the fitting dataset) to train, test, and select random forest regression models. The target variables were the concentration of total, available, organic, and inorganic P forms in the topsoil. The predictive features in the fitting dataset were latitude, longitude, sand, silt, and clay fractions, mean annual precipitation and temperature, pH, elevation and slope, total N, total organic C, and reference soil

group. Using the selected models and a compiled spatially explicit dataset (0.5° x 0.5° lat-long) containing 2749 grid cells with estimated values of the predictive features found in the fitting dataset, we constructed estimated maps of the target P forms for the pan-Amazon region.



### 4.1 Soil Phosphorus maps

The mean concentration of total P in the fitting dataset (284.13 mg kg$^{-1}$) shows that the pan-Amazon region is poor in phosphorus when compared with a global mean of 570 mg kg$^{-1}$ (He et al., 2021). The Amazonian foreland basins (Fig. S6) are marked by relatively higher values of total P. These are Cenozoic sedimentary basins in western Amazon and are under the influence of the ongoing Andean orogenesis uplifting (Val et al., 2021 and citations therein) during the last 10 million years. The regional atmospheric circulation, influenced by orogenic effects, causes high precipitation rates along the Andes foothills

in western amazon (Bookhagen and Strecker, 2008). The transport of primary material enriched with P from the Andes foothills through Amazonian foreland basins and through the lowland catchments of the Amazonas, Solimões, Juruá, and Purús Rivers in the central Amazon (Solimões and Amazon sedimentary basins, Fig S6) results in a relatively higher total P in these regions (Wittmann at al., 2011, Quesada et al. 2010). In contrast, the sedimentary basins in the lowlands of eastern Amazon that are characterized by approximately 20 millions of years of tectonic stability under the influence of the weathered crystalline

outcrops of the Proterozoic rocks of the Brazilian and Guinan shields (Quesada et al., 2010). The generated map of total P (Fig. 1) resembles the predicted patterns found in the literature (Val et al., 2021) and in comparison with the more recent map of total P (He et al., 2021) our map presented a better correlation coefficient with observed data. However, it is important to note that despite the fact that the dataset used to train our random forest models was a subset of the data used to train an test the random forest model used by He et al. (2021), the later was trained using global data aiming to the production of a global

map, indicating that training models to restricted areas could be important to optimize the accuracy and predictive power. The difference between the predicted map of total P (Fig. 1) and the sum of the compounding P forms (Fig. 7) is a proxy for the occluded P form in most of the depicted areas with the exception to the southeast-northwest band along the Andes, where the abundance of P-rich primary material is expected to be higher than in the older soils of the Amazon lowlands due to the high weathering intensity and transport of parental material associated with the environmental conditions in the border between the

Andes and the Amazon. The spatial distribution of the occluded P (Fig. 7), organic P (Fig. 3), and inorganic P (Fig. 4) followed the pattern observed in the total P map (Fig. 1), indicating that the sizes of these P pools are related with the concentration of total P (He et al., 2023).

### 4.2 Important variables and partial dependence

The analysis presented here showed the dependence of all target P forms, especially total P, on total N (Fig S8). Considering

the theory about the fate of these elements in soils (Walker and Syers, 1976; Lambers et al., 2008), one possible explanation to this observed link comes from the development stage of the soils in the region, that are originated from old lithologies (minimum age > 1.5 million years) and characterizted by millions of years of geological stability and countinuous weathering (Val et al., 2021). Meaning that the younger soils in the fitting dataset are old if compared with volcanic soils in the common chronosequences studied in Hawaii, for example (Crews et al., 1995). Quesada et al. (2010) already identified a positive

correlation between total P and total N. In general, higher values of total P concentration  and total N were found in the younger reference soil groups in the fitting dataset, with the concentration of both N and P decreasing with soil age, indicating that the





correlation between total N and total P could be explained by the gradient of soil particle specific surface area, surface charge densities, and organic matter adsorption (Quesada et al., 2010). Another concurrent explanation could be the possible control of P on N fixation  (Reed et al., 2013, but see Wong et al., 2020). In one hand, for younger and less weathered soils the main

source of N is biological fixation and for P it is weathering and/or deposition of primary minerals rich in P (Lambers, 2022). On the other hand, older and weathered soils are characterized by the continuous loss of N and P and the accumulation of P as occluded forms (Crews et al., 1995), with the organic forms of both elements exerting a strong control on the nutrient availability to plants (da Silva et al., 2022). The predominant low nutritional status of the soils in the fitting dataset and the positive relation between total N and total P indicates that in the very old and weathered soils of the pan-Amazon region, the

low availability of P may be a limiting factor for biological N fixation (Liese et al., 2017; Van Langenhove et al., 2021; Zhang et al., 2022).

Besides total N, other variables also presented high values of MDA for the target P forms. The variable pH has high values of MDA for all target P forms with the exception of the inorganic P. The partial dependence of total P on pH (Fig. 8) indicates positive relation between pH and total P. This is expected because in the sampled sites, in general, a greater concentration of

total P is associated with younger soils that have a higher sum of bases and less exchangeable Al, which in turn are associated with higher pH values (Quesada et al. 2010). Mean annual precipitation presented high values of importance for the prediction of available P (Fig. 6) and a negative relation with available P (Fig. S10). The strong dependence of available P on precipitation is explained by a few sites that are characterized by young soils (Umbrisols and Cambisols) that have an overall greater total P concentration and present MAP lower than 1000 mm/y. In these sites, the low precipitation rates can contribute to a low-

intensity transport of water-soluble fractions of P. Unsurprisingly, mean annual temperature presented high values of importance for the prediction of organic P (Fig. 6) and a negative correlation with the organic P form (Fig S12). Temperature is one of the main drivers of organic matter decomposition (Howe and Smith, 2021), and thus, it is reasonable to expect that under lower decomposition rates, the concentrations of P related to organic matter should be relatively higher. The sand fraction presented high values of MDA for the prediction of the inorganic P form (Fig. 6), and the partial dependence analysis (Fig.

S14) shows a negative relation between sand fraction and inorganic P. The reduced adsorption capacity in the sandy soils of the fitting dataset can explain the negative relationship with the inorganic P form (Quesada et al. 2010; Osman, 2018). The relations between the target P forms (and the fractional components of the Hedley sequential extraction) and the most important features of the fitting dataset are analysed and discussed extensively in the study of Quesada et al. (2010).

### 4.3 Uncertainties related to the fitting and predictive datasets

The reference soil groups presented low importance values for predicting the target P forms (Fig. S15). This indicates that the spatial extrapolation of P forms made from Hedley fractionation data based on reference soil groups can lead to a wrong interpretation of the concentrations and proportions of P forms in soils. The low number of observations of Hedley fractionation in different soil types (Yang and Post, 2011) leads to a high coefficient of variation of Hedley P fractions for each reference soil group. Moreover, there are a few soil classes that are under-represented in the fitting dataset (Table S4 and Table 1 in

Quesada et al., 2011). In general, this is the case for sites with higher elevations characterized by the occurrence of younger
soils like Leptosols and Andosols (Figs. S16 and Table S4). Important intervals in the upper and/or lower ranges of some
features related to elevation are absent in the training data, as shown by the dissimilarity index analysis (Fig. S3), reducing the
suitable area to predictions using the random forest models. Our random forest regression models presented a good level of
accuracy during the training phase. Nonetheless, the results could be improved by a greater number of in situ observations of
soil P fractions and other edaphic and climatic variables in the Amazon region, especially in high elevational areas where
observations are lacking, with notable exeptions (Wilcke et al. 2019). Finally, it is important to note that our P maps rely on
raster datasets used to predict the P pools, which are subject to their own uncertainties and inconsistencies. The improvement
of these maps is also important to the precision and accuracy of mapping exercises as we present here.

## 5 Outlook

The maps presented here provide information about P stocks and their distribution in soils. They are available for developing
and evaluating DGVMs that seek to include or improve the representation of P cycling and P limitation on primary productivity
in the pan-Amazon region. Additionally, the data presented here can be useful for correlational spatial studies that promote
new hypothesis linking P availability and vegetation structure and function that could be tested in the study area. However,
caution is needed with regard to the temporal variability of these P forms. The P forms have different residence times ranging
from hours to millennia and are subject to a complex set of interactions with biotic, edaphic, and climatic environmental
attributes over time. In this scenario, the presented maps can be useful to define initial conditions to dynamic, process-oriented
models, for the simulation of P cycling in soils (Helfenstein et al. 2018). Random forests are recognized for their power in
prediction exercises; however, the methods to unveil the underlying drivers and mechanisms relating target variables and
explanatory feaures are still under development (Lucas, 2020; Simon et al., 2023). Moreover, the dataset used to fit the random
forest models is subject to a lack of observations in elevated regions and have their methodological limitations and
uncertainties. Nonetheless, the method applied here to estimate the spatial distribution of P forms in the topsoil of the pan-
Amazon region relies on the relationships that maximize the amount of information that each predictive variable (i.e., feature
related to the soil P content at the landscape scale) can contribute to the random forest regression models. Thus, our approach
can partially overcome the lack of certainty in the Hedley fractionation P forms/soil classification relationship by applying a
nonparametric method based on statistical learning to predict the P foms in the soils of the pan-Amazon region.

## Data availability

The phosphorus maps for the pan-Amazon region can be downloaded from https://doi.org/doi:10.25824/redu/FROESE
(Darela-Filho and Lapola, 2023).

**Code availability**

The source code and input data used to produce the maps are available at jpdarela/Reference_phosphorus_maps_pan-Amazon (github.com)

**Author contributions**

JPDF, AR, DML, KF, LCCH, and TR conceptualized the experiment and JPDF caried it out. JPDF developed the code and the visualizations.CAQ was responsible for data sampling and curation. JPDF prepared the manuscript with contributions from
all co-authors.

**Competing interests**

The authors declare that they have no conflict of interest.

**Acknowledgements**

We are thankful to Andy Krause for the insightful comments on the manuscript. TR would like to thank for the financial
support of the International Graduate School of Science and Engineering (IGSSE-TUM) through the grant associated with the PhosForest project and the financial support of the Bayerische Staatskanzlei (Bavarian State Chancellery) through the grant associated with the Amazon-FLUX project. LFL would like to acknowledge the Bavarian State Chancellery (Project Amazon-FLUX) for financial support. DML would like to acknowledge the Sao Paulo Research Foundation - FAPESP (grant nº 2015/02537-7) and the Brazil's National Council on Research and Technological Development - CNPq (grant nº 309074/2021-
5). JPDF would like to thank for the financial support of the Sao Paulo Research Foundation - FAPESP (grant nº 2017/00005-3 and grant nº 2019/08194-5).

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

**Figures**

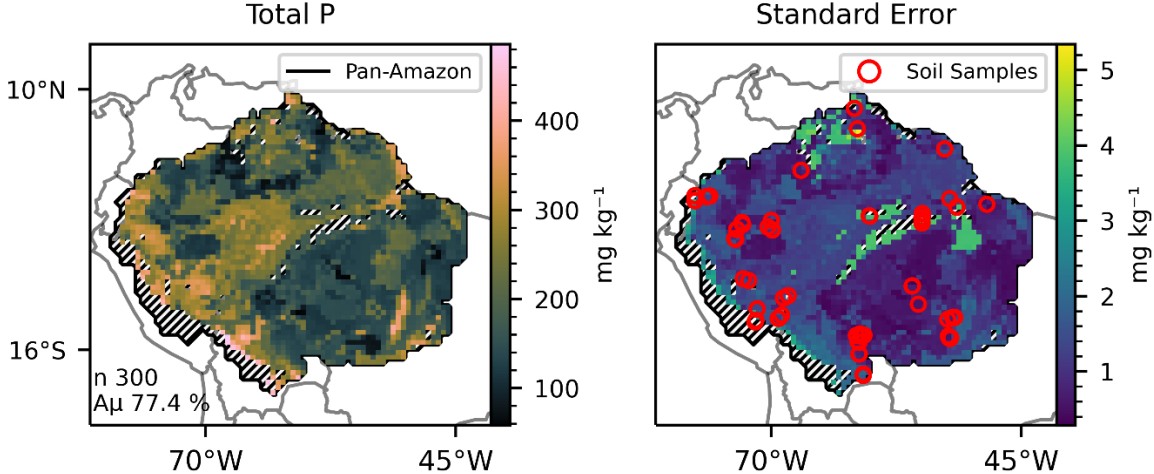

**Figure 1** Left, mean total P predicted by 300 selected random forest models with mean accuracy of 77.4 % at the training/testing phase. Right: Standard Error of the 300 predicted maps. The hatched areas mark the regions where the Dissimilarity Index (DI) 
presented values greater than the sum of the third quartile with the inter-quartile range. Red circles mark the sites with data collections for the fitting dataset.

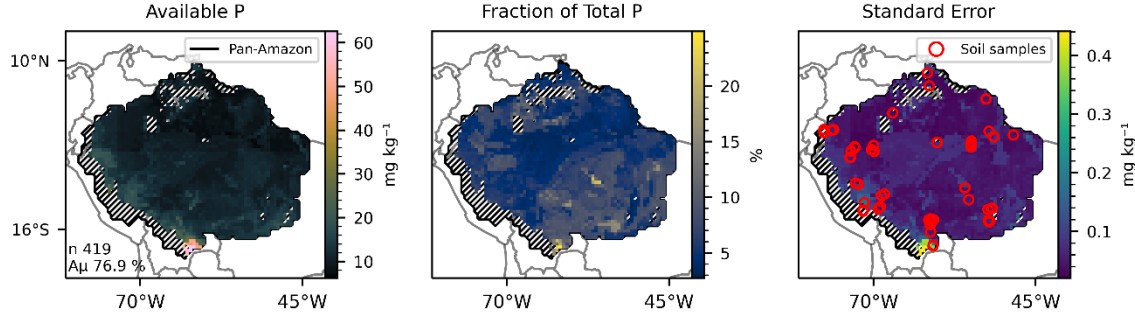

**Figure 2:** Left, mean available P concentration predicted by the 419 selected Random Forest models with mean accuracy of 76.9 % at the training phase. Middle: Fraction of mean available P as percentage of the predicted total P concentration. Right: Standard 
Error of the 419 predicted maps. The hatched areas mark the regions where the Dissimilarity Index (DI) presented values greater than the sum of the third quartile with the inter-quartile range. Red circles mark the points of data collection of the fitting dataset.
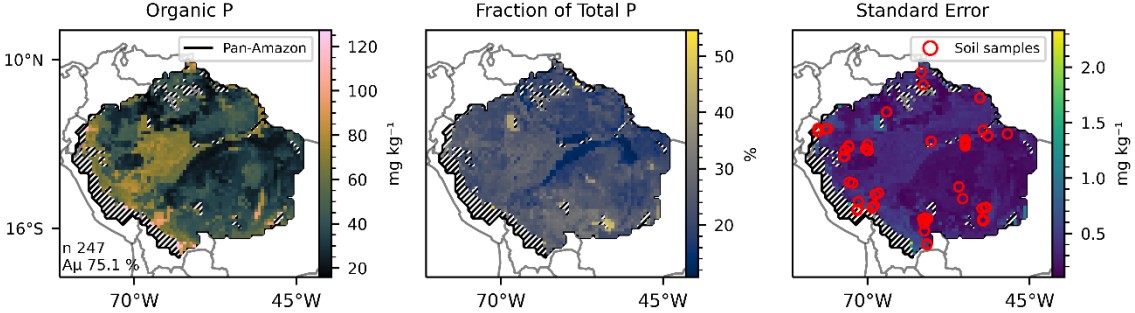

**Figure 3:** Left: mean Organic P predicted by 247 selected Random Forest models with mean accuracy of 75.1 %. Middle: Fraction of the mean Total P represented by the mean Organic P. Right: Standard Error of the 247 predicted maps. The hatched areas mark the regions where the Dissimilarity Index (DI) presented values greater than the sum of the third quartile with the inter-quartile range. Red circles mark the points of data collection for the fitting dataset.

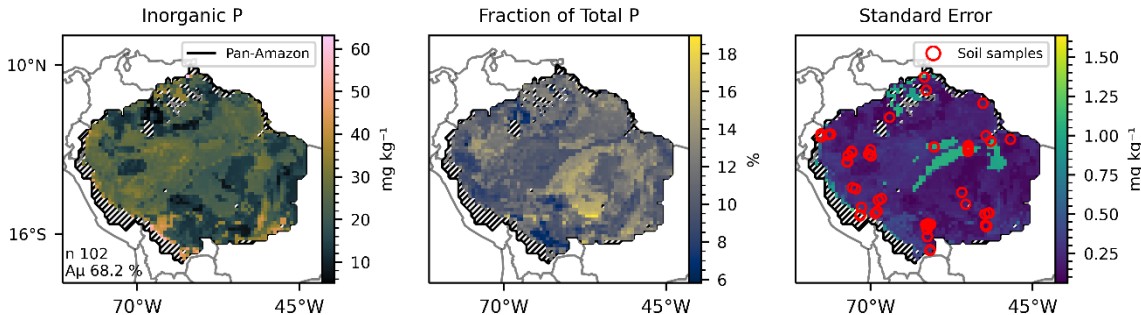

**Figure 4:** Left: mean Inorganic P predicted by 102 selected Random Forest models with a mean accuracy of 68.2 % in the training/testing phase. Middle: Fraction of the mean Total P represented by the mean Inorganic P. Right: Standard Error of the 102 predicted maps. The hatched areas mark the regions where the Dissimilarity Index (DI) presented values greater than the sum of the third quartile with the inter-quartile range. Red circles mark the points of data collection for the fitting dataset.

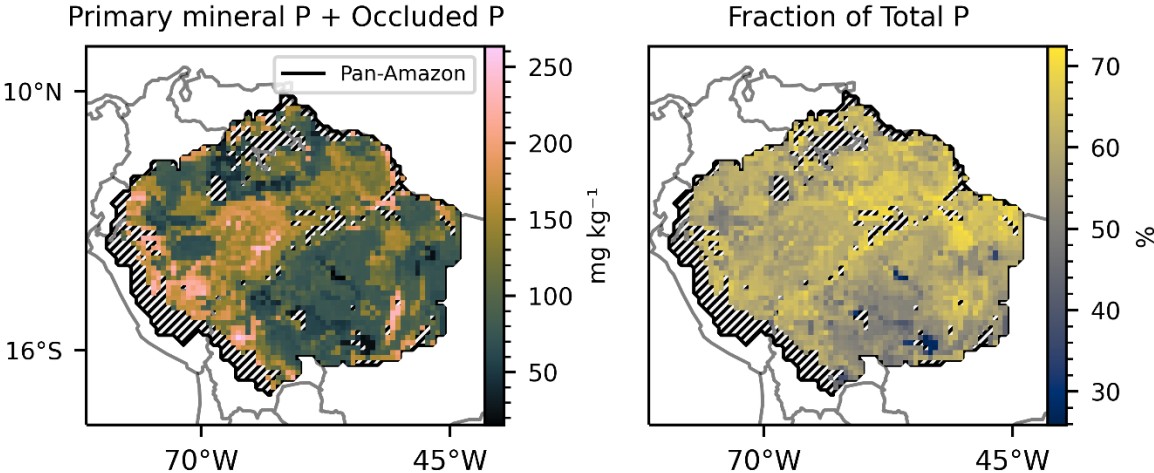

**Figure 5: Left: Map generated by the subtraction between mean total P and the sum of the remaining predicted P forms. Right: percentage of the mean total P represented by mineral and occluded P, depicted on the left. The hatched areas mark the regions where the Dissimilarity Index (DI) presented values greater than the sum of the third quartile with the inter-quartile range in the total, available, organic, and inorganic P forms maps.**


Earth System
Open Access Science
Data Discussions



**Figure 6: Variable's (excluding soil reference groups) permutation importance – or MDA (Mean Decrease in Accuracy), distribution of means for the set of RF models selected for each P form (Table 3). Positive (negative) values of MDA indicates that the 'exclusion' of the variable decrease (increase) the RF model accuracy. Higher values of MDA indicate higher variable importance. Each selected model was permuted 120 times. The internal variability (Standard Deviation of MDA) of each model is not presented. Abbreviations: TN: total nitrogen; TOC: total organic carbon; MAP/MAT: mean annual precipitation/temperature; lat: latitude; lon: longitude. See Table 1 for variable units.**






**Tables**

**Table 1: Measured variables used in this study. The P pools sizes are based on the grouping of different fractions sequentially extracted of soil samples (see Table 2). All soil measurements were collected in the 0-30cm soil profile.**

| Feature | Units |
| --- | --- |
| Latitude (lat) | Decimal Degrees North (WGS84) |
| Longitude (lon) | Decimal Degrees East (WGS84) |
| Reference soil group | WRB major reference soil groups |
| Sand, silt, and clay | % |
| Slope | % |
| Elevation | Meters (m) |
| MAT (Mean annual temperature) | °C |
| MAP (Mean annual precipitation) | mm year$^{-1}$ |
| Topsoil pH in water | -log(H+) |
| TOC (Total organic carbon) | % |
| TN (Total Nitrogen) | % |
| Inorganic P | mg kg$^{-1}$ |
| Organic P | mg kg$^{-1}$ |
| Available P | mg kg$^{-1}$ |
| Total P | mg kg$^{-1}$ |

**Table 2: Ecological P forms modelled in this study and the respective fractions of P obtained with the methods described in Hedley**
**et al., (1982); Quesada et al., (2010) and Hou et al., (2018). The total P was extracted from a replicate sample using the method of Tiessen and Moir (Carter and Gregorich, 2008). Pi is inorganic P fraction. Po is organic P fraction as in Hou et al. (2018).**

| Forms of P modelled in this study | Hedley fractions |
| --- | --- |
| Available P | Resin P, NaHCO$_3$ Pi |
| Organic P | NaHCO$_3$ Po, NaOH Po |
| Inorganic P | NaOH Pi |
| Total P | H$_3$SO$_4$ + H$_2$O$_2$ in a replicate sample |





**Table 3: Threshold values for model selection (see the main text, sect. 2.1), number of selected models and mean accuracy (Aμ) obtained for each P fraction.**

| P form | Min. $A$ value (%) | Min. $R^2$ in cross-validation | Number of selected models | Aμ of selected models (%) | Mean cross-validation $R^2$ of selected models |
|---|---|---|---|---|---|
| Available P | 75 | 0.55 | 419 | 76.86 | 0.59 |
| Organic P | 73 | 0.55 | 247 | 75.14 | 0.67 |
| Inorganic P | 65 | 0.55 | 102 | 68.23 | 0.57 |
| Total P | 75.8 | 0.55 | 300 | 77.37 | 0.7 |