# Peer review of "Reference maps of soil phosphorus for the pan-Amazon region"

_Earth System Science Data, 2023_

## Author Response (AR1)

Dear Editor,

We sincerely thank you for considering our data description paper for publication in Earth System Science Data. We also thank the anonymous referees for their time, insightful comments, and recommendations, which helped us improve our manuscript. After carefully considering all feedback, we believe that the dataset and the data description paper have significantly improved.

In this revised version, we have reconstructed the dataset with a higher spatial resolution (5 arcminutes). Additionally, some of the geospatial datasets used in the methodology have been replaced with more recent products. We have also updated the manuscript to incorporate the suggestions of the anonymous referees. Specifically, we have clarified our model selection approach and addressed other methodological aspects that were pointed out.

Below, we reproduce the comments of Referees #1 and #2, *in italics*, followed by our responses (**Author's Response**).

**Referee #1**

***Resolution Rationale:*** *The manuscript lacks a clear justification for selecting a 0.5-degree resolution for mapping. Providing insight into the reasoning behind this choice would enhance the manuscript's robustness.*

**Author's Response**: We thank the referee for this comment. Our initial choice of resolution was influenced by the primary purpose of these maps, which was to support the parameterization and evaluation of land surface/vegetation models. These models are computationally intensive and generally use a 0.5-degree resolution. However, we acknowledge that other uses of the reference maps, as proposed in the manuscript (Lines 385 and 395), could benefit from a finer resolution. Therefore, we are pleased to offer the set of reference maps at a 5-arcminute resolution. Please, note that the line numbers refer to the .pdf file of the reviewed manuscript with tracked changes.

During this process, we revisited the pre-processing of the predictive data and incorporated more recent datasets that have been published. Now, all variables, apart from temperature, precipitation, and elevation, are sourced from SoilGrids 2.0 [International Soil Reference and Information Centre] (Poggio et al., 2021).

We have added the following text in Section 2.2 to clarify the rationale behind our chosen resolution.

> "All spatial raster datasets were downloaded from the sources and used in the resolution of 5 arcminutes. While it is possible to obtain the data in a finer resolution, the primary intent of the maps presented here is to parametrize and benchmark land surface models that simulate terrestrial vegetation. Thus, we opted to produce the maps in the resolution of 5 arcminutes. In this resolution, the maps can be easily aggregated to satisfy the needs of land surface modelling at the same time enabling other possible uses of the reference maps that requires a higher resolution."

*Methodological Explanation: Certain methodological aspects, such as the approach to Random Forests model selection and the use of $10^5$ Random Forests models for 108 observations, require more detailed explanation and justification. The reason for excluding primary mineral P and the occluded P forms were not solid.*

**Author's Response**: We thank the referee for this comment. Regarding the model selection approach, we added the following text in the section 2.3:

> "We chose this selection approach due to the inherent stochasticity in both the train/test split phase and the training of Random Forest models. In the former, samples from the dataset are randomly assigned to either the training or testing sets. In the latter, stochasticity arises from two factors: (i) bootstrap sampling, where each decision tree is trained on a random sample (with replacement) from the dataset, and (ii) feature randomness during decision tree construction. Unlike standard decision tree construction, which uses the feature that provides the most information gain for a split (or tree branch), Random Forests build each tree based on a random subset of features from the training data. Therefore, by selecting a group of models from a pool, we can capture the inherent stochasticity in the models while choosing the most accurate ones."

Upon revisiting our model selection, we decided to include the Random Forest models trained on the occluded P form in the production of the reference maps. Although the number of selected models and their accuracy were lower compared to other P forms, we decided to include them due to the positive feedback from the reviewers. As mentioned, the initial choice to not include the occluded P in the model fitting was based on the lower number of selected models for this P form. Nonetheless, due to the significance of occluded P to the understanding of ecological processes in the pan-Amazon region, we decided to include after the review.

Unfortunately, this was not possible for primary mineral P. In our model selection approach, all models trained to predict primary mineral P demonstrated very low accuracy values. We believe that this is caused by the trace amounts of mineral P form (Calcium bound P) found in most of the samples in the fitting dataset. Which in its turn, is related to the lack of observations in the most P rich sites. Additionally, the trace amounts of Calcium bound P found in the samples are explained by the elevated pH and advanced weathering stages of these soils. We have added an explanation for the exclusion of mineral P from model fitting and selection in Section 2.3:

> "In our model selection approach, the models fitted for the primary mineral phosphorus (P) form demonstrated very low accuracy, with the best cases achieving only 15%. Consequently, we did not include the primary mineral P form in the initial part of the analysis, which involved model fitting. We attribute this to the extremely low values of Calcium-bound P observed in most samples in the fitting dataset. The majority of data in the fitting dataset were collected from sites with old, well-weathered, and acidic soils, characterized by trace amounts of Calcium-bound P. Furthermore, we concluded that our set of predictive variables, considering both the geographical context and the distribution of sampled sites, was insufficient to generate an inference model for primary mineral P. To address this issue, we estimated the size of the primary P pool by subtracting the combined total of available, organic, inorganic, and occluded P forms from the estimated total P. We interpret this as an

indication that the information from the set of variables in Table 1 is insufficient to generate predictions for the primary P form."

*Temporal Representativeness: The temporal scope of the soil P estimates needs to be clarified and discussed. It would be beneficial to address the use of data collected at different periods and its potential impact on the results, especially in the context of changing soil conditions.*

**Author's Response:** The comment raises a crucial point, and we thank the reviewer for pointing it out. The creation of the P reference maps assumes that the size of the P forms pools remains constant during the sampling process. This significant assumption was not mentioned in the previous version of the manuscript. Given the timescales of P transformations in soils, we believe this to be a reasonable assumption. Unfortunately, the challenges associated with data collection in the Amazon are unparalleled in terms of available human and economic resources. This is an impediment to continuous survey campaigns aimed at consecutive collections of data in the region. It is also beyond the scope of our study to investigate the dynamics of P forms in soils on this spatial scale. We have now included this information at the end of subsection 2.1 in the Material and Methods section:

> "The limited number of samples and the spatial gaps in the dataset used for fitting are understandable, considering the mobility challenges in the region. Similarly, the sample collection is temporally heterogeneous due to these constraints, limiting opportunities for repeated sampling over extended periods (Carvalho et al. 2023). The reference maps constructed here are based on the assumption that the size of the P forms pools in soils remain stable during sampling. This implies that the transformation of some P forms into others does not significantly alter the size of the P form pools during data collection. Given the geological timescales of P's biogeochemical cycling, we consider this a reasonable assumption. However, understanding the dynamics of P forms in soil falls outside the scope of this study."

In section 4.3, we propose a potential use for the reference maps in studies investigating the dynamics of P forms in soils:

> "The P forms have different residence times ranging from hours to millennia and are subject to a complex set of interactions with biotic, edaphic, and climatic environmental attributes over time. In this scenario, the presented maps can be useful to define initial conditions to dynamic, process-oriented models, for the simulation of P cycling in soils (Helfenstein et al. 2018)."

*Additionally, the manuscript could be strengthened by:*

**High-Resolution Covariate Exploration:** *Given that many relevant soil P covariates are available at finer spatial grids, discussing the potential benefits of reproducing this study with higher spatial resolution information would enhance the value of the presented soil P data.*

**Author's Response:** We thank the referee for this comment. As previously mentioned, we have revisited the pipeline for creating the reference maps and rebuilt it with a finer resolution

(5 arcminutes). In addition, we have replaced the predictive geospatial data on soil physiochemical attributes with more recent products.

***Sensitivity to Spatial Support:*** *An exploration of how soil P predictions might vary with different spatial support levels would provide valuable insights into the robustness of the results.*

**Author's Response:** The feedback from the referee is appreciated. We acknowledge that an analysis of this nature would require the exclusion or aggregation of some sampled points for the training of the Random Forest models. Given the limited number of samples, we chose to use a multivariate dissimilarity index. This approach allows us to avoid applying the models to data outside the training range, while testing the generality of the selected models using a Monte-Carlo cross-validation. In our view, this is the most effective approach, given the constraints imposed by the small number of samples and the characteristics of the Random Forest algorithm.

While we agree that some variables in the fitting dataset have different spatial supports (for example, pixels for climatic data and soil cores for soil data), it's important to note that the Random Forest algorithm is not a geostatistical interpolation technique. Therefore, its requirements and assumptions for application in a geospatial context differ.

*Finally, on a minor note, it's important to consistently capitalize "Random Forests" throughout the manuscript, as it is the name of the algorithm.*

**Author's Response:** We thank the referee for noticing it. We have now ensured that the term "Random Forest" is consistently capitalized throughout the manuscript.

**Referee #2**

***Lack of Innovation in the Fitting Dataset:*** *It is noted that your data sources are primarily from Hou et al., 2018, and Quesada et al. (2020). However, it is apparent from Supplementary Figure S6 that there is a lack of observation data in the central extensive Solimoes Basin. This gap in the dataset should have been addressed or justified in the manuscript.*

**Author's Response:** We concur with the referee regarding the sparse nature of the sampled data. The limitations inherent in data collection across the pan-Amazon region are due to the limited mobility options. Many of the sampled sites can only be accessed through lengthy journeys along rivers and trails in the heart of the forest (Carvalho et al. 2023).

In our view, the world's largest tropical forest has not received the attention it deserves, despite the tremendous efforts of scientists who spend considerable amounts of time and risk their lives to collect data in the Amazon wilderness. Furthermore, our study's primary objective was to use available data with an alternative statistical method, to overcome the

challenges encountered by other methods that used the same data to map P forms in the region.

We have addressed this issue by adding a paragraph to Section 2.1 in the manuscript. We kindly ask to the referee #2 to check our response to the issue "Temporal Representativeness", pointed by the referee #1.

***Resolution Insufficiency:*** *The manuscript does not provide a clear justification for selecting a 0.5-degree resolution for mapping. Given the availability of higher-resolution climate data and other relevant soil physicochemical properties at finer resolutions, the choice of such a coarse resolution for a relatively small study area requires further explanation.*

**Author's Response:** We value the referee's suggestion and have taken it into consideration. Our original choice for a half-degree resolution was based on the anticipated use of the reference maps for the parameterization and evaluation of land surface/vegetation models. We agree that, given the relatively small study area, the initial choice of spatial resolution was not optimal. To address this issue, we have reconstructed the reference maps at a finer spatial resolution of five arcminutes. Additionally, we have incorporated more recent datasets of soil physiochemical properties. All geospatial datasets with soil properties are now sourced from SoilGrids 2.0 [International Soil Reference and Information Centre] (Poggio et al., 2021). We have added a sentence in Section 2.2 to clarify the rationale behind our chosen resolution. The referee #2 can find it in the response to the first point addressed by the referee #1, "Resolution Rationale".

***Lack of Methodological Innovation:*** *The manuscript mentions suboptimal model performance in high-altitude regions. It would be beneficial to explore the possibility of stratifying the analysis, perhaps by altitude or dominant vegetation types. This would demonstrate the flexibility of the Random Forest methodology. Additionally, if there is a lack of representative data in the central part of the study area, consider leveraging transfer learning techniques with data from other similar terrains globally.*

**Author's Response:** We acknowledge and appreciate the referee's point of view on this matter. However, we would like to clarify that we did not claim that we observed suboptimal model performance at high altitudes. The model's performance was evaluated using appropriate metrics (accuracy, $R^2$, MAE) and a Monte-Carlo cross-validation. The limited number of observations in these high elevation environments prevented us from applying the fitted models in areas where the predictive variables showed high multivariate dissimilarity between the fitting dataset and the predictive dataset.

The exclusion of some areas in the maps is related to a well-known limitation of the method applied. The Random Forest algorithm can be inaccurate when data outside the ranges used in the training phase is used for testing or prediction. We observed, *a posteriori*, that the areas excluded after using the Dissimilarity Index (DI) are characterized by high elevation.

Due to the small number of sampled points, especially in the most elevated areas, we believe that a stratified analysis is not an optimal approach as it would require us to train the models

on even smaller subsets of data. Moreover, including a categorical variable defining elevation would be redundant, because elevation is already in the dataset.

Regarding the dominant vegetation type, most of the sampled points in the fitting dataset are in forests. A categorical variable defining vegetation type would be severely unbalanced and uninformative. Finally, as shown using the DI, despite the small number of observations in the Solimões basin, the region is well represented in terms of all variables when compared to the most elevated areas.

Nonetheless, new data collection campaigns throughout the study area could improve future studies.

*Insufficient Model Validation: The manuscript could benefit from a comparison of simulation results with other relevant data sources such as the World Soil Information database, especially when using inputs from various soil databases. This would enhance the robustness of your findings.*

**Author's Response:** We appreciate the referee's comment, but we find it unclear. The World Soil Information database does not provide maps of P forms in soil that could be used to evaluate our maps. This database is now the source for a subset of the covariates used in constructing the P maps at the new resolution of five arcminutes. As the World Soil Information database represents the state-of-the-art in terms of geospatial soil datasets, we consider that the application of the models using outdated soil datasets would not be of great benefit.

*Temporal Representativeness: The temporal scope of your soil P estimates should be clarified and discussed, especially considering data collected at different time periods. Exploring the temporal variation of soil P and its relationship with climate change would add depth to your study.*

**Author's Response:** We kindly ask Referee #2 to refer to our response to the same issue raised by Referee #1. In that response, we clarified the caveats related to the temporal scope of data collection and stated the main assumption that the size of the P pools did not change significantly during the data acquisition period.

While we agree that the temporal variability of P in soil and its relationship with climate change is an important topic, discussing the impacts of climate change on the dynamics of P forms would require a different methodological approach. Therefore, we believe that this topic is outside the scope of this study.

*Sensitivity Analysis: It is crucial to conduct sensitivity analyses to assess how soil P predictions might vary with different spatial support levels. This would provide valuable insights into the robustness of your results.*

**Author's Response:** We thank the referee for this comment. We kindly ask Referee #2 to refer to our response to the same issue raised by Referee #1 regarding **sensitivity to spatial support**.

**References**

Carvalho, R. L., Resende, A. F., Barlow, J., Franca, F. M., Moura, M. R., Maciel, R., Alves-Martins, F., Shutt, J., Nunes, C. A., Elias, F., Silveira, J. M., Stegmann, L., Baccaro, F. B., Juen, L., Schietti, J., Aragao, L., Berenguer, E., Castello, L., Costa, F. R. C., Guedes, M. L., Leal, C. G., Lees, A. C., Isaac, V., Nascimento, R. O., Phillips, O. L., Schmidt, F. A., Ter Steege, H., Vaz-de-Mello, F., Venticinque, E. M., Vieira, I. C. G., Zuanon, J., Synergize, C., and Ferreira, J.: Pervasive Gaps in Amazonian Ecological Research, Curr Biol, 33, 3495-3504 e3494, https://doi.org/10.1016/j.cub.2023.06.077, 2023.

Poggio, L., de Sousa, L. M., Batjes, N. H., Heuvelink, G. B. M., Kempen, B., Ribeiro, E., and Rossiter, D.: Soilgrids 2.0: Producing Soil Information for the Globe with Quantified Spatial Uncertainty, SOIL, 7, 217-240, https://doi.org/10.5194/soil-7-217-2021, 2021.

Helfenstein, J., Tamburini, F., von Sperber, C., Massey, M. S., Pistocchi, C., Chadwick, O. A., Vitousek, P. M., Kretzschmar, R., and Frossard, E.: Combining Spectroscopic and Isotopic Techniques Gives a Dynamic View of Phosphorus Cycling in Soil, Nat Commun, 9, 3226, https://doi.org/10.1038/s41467-018-05731-2, 2018.